# The Mean Unfulfilled Lifespan (MUL): A new indicator of the impact of mortality shocks on the individual lifespan, with application to mortality reversals induced by COVID-19

**Patrick Heuveline**⬚*

California Center for Population Research (CCPR), University of California, Los Angeles (UCLA), Los Angeles, CA, United States of America

* heuveline@soc.ucla.edu

**Data Availability Statement:** All raw data are from open online sources cited in the manuscript. Additional data files prepared from those have been

## Abstract

Declines in period life expectancy at birth (PLEB) provide seemingly intuitive indicators of the impact of a cause of death on the individual lifespan. Derived under the assumption that future mortality conditions will remain indefinitely those observed during a reference period, however, their intuitive interpretation becomes problematic when period conditions reflect a temporary mortality "shock", resulting from a natural disaster or the diffusion of a new epidemic in the population for instance. Rather than to make assumptions about future mortality, I propose measuring the difference between a period average age at death and the average expected age at death of the same individuals (death cohort): the Mean Unfulfilled Lifespan (MUL). For fine-grained tracking of the mortality impact of an epidemic, I also provide an empirical shortcut to MUL estimation for small areas or short periods. For illustration, quarterly MUL values in 2020 are derived from estimates of COVID-19 deaths that might substantially underestimate overall mortality change in affected populations. These results nonetheless illustrate how MUL tracks the mortality impact of the pandemic in several national and sub-national populations. Using a seven-day rolling window, the empirical shortcut suggests MUL peaked at 6.43 years in Lombardy, 8.91 years in New Jersey, and 6.24 years in Mexico City for instance. Sensitivity analyses are presented, but in the case of COVID-19, the main uncertainty remains the potential gap between reported COVID-19 deaths and actual increases in the number of deaths induced by the pandemic in some of the most affected countries. Using actual number of deaths rather than reported COVID-19 deaths may increase seven-day MUL from 6.24 to 8.96 years in Mexico City and from 2.67 to 5.49 years in Lima for instance. In Guayas (Ecuador), MUL is estimated to have reached 12.7 years for the entire month of April 2020.

## Introduction

For months, the numbers of deaths from the novel coronavirus disease 2019 (COVID-19) have become part of the daily news cycle the world over. Even when related to the population size in deaths per capita ratios, however, these numbers do not really provide any intuition for

placed in the GitHub repository (https://github.com/heuveline/ind-cov-mort/).

**Funding:** The author received no specific funding for this work, but benefited from the research environment of the California Center for Population Research at UCLA (CCPR), which receives core support (P2C-HD041022) from the Eunice Kennedy Shriver National Institute of Child Health and Human Development (NICHD). The funders had no role in study design, data collection and analysis, decision to publish, or preparation of the manuscript.

**Competing interests:** The authors have declared that no competing interests exist.

the magnitude nor the dynamics of the pandemic. Quite useful for between-population comparisons, the age standardization of these ratios does not make them more easily interpretable.

The period life expectancy at birth (PLEB) is probably the most readily interpretable of the period indicators mortality. Translating a number of deaths from a given cause into its impact on PLEB involves multiple steps but is fairly straightforward [1, 2]. Unfortunately, the intuitive appeal of the PLEB, its interpretation as a measure of the individual lifespan, derives from the assumption that period mortality conditions will continue to prevail indefinitely. Mortality conditions are always changing, but when these changes are relatively steady, changes in PLEB remain interpretable as changes in individual longevity [3]. When declines in PLEB are induced by a relatively rapid and likely temporary increase in mortality, such as currently experienced with the COVID-19 pandemic, however, they become hardly interpretable as indicators of changes in the individual lifespan [4].

The interpretation is even more problematic when PLEB or changes therein are estimated for smaller populations and shorter periods of time. Tracking the pandemic at a finer-grained geographical and temporal scale undoubtedly provides better insights on the pandemic than annual, national averages [5–7]. But while the assumption underlying an annual PLEB estimate—that mortality conditions of a given year will be repeated year after year in the future—may seem unlikely, the seasonality of mortality makes the indefinite repetition of the mortality conditions in any fraction of a year plainly impossible. Referring to mortality conditions not only in a short period but also in a small area, these "pseudo" PLEB estimates [8] build on assumptions akin to a Groundhog-Day [9] time loop repeating itself in a small area from which individuals are unable to leave.

Aware of the pitfalls of interpreting changes in PLEB at the individual level, demographers may only provide them as macro-level measures of mortality shocks across time and place, but their widespread misinterpretation during the COVID-19 pandemic reveals the lack of a both interpretable and scalable, over space and time, measure of what changing mortality conditions mean for the individual lifespan. This article proposes such a measure, the Mean Unfulfilled Lifespan (MUL). Making no assumption about future mortality, the MUL translates past changes in mortality into an average difference in the length of lived lives. That difference is obtained by comparing the actual average age at death during a given period and the expected ages at death of the same individuals (death cohort) in the absence of mortality changes, whether induced by a specific cause of death or by an event affecting multiple causes of death. The MUL remains interpretable for populations of any size and for periods of any length, and can be estimated from data on excess deaths or cause-specific deaths, as illustrated here with quarterly COVID-19 mortality data. The MUL also equals the product of (1) the proportion of deaths in a population and given period from a cause or due to an event of interest and (2) a weighted average of counterfactual life expectancies by age and sex in the absence of that cause or event of interest, with weights provided by the distribution of deaths from that cause or due to that event of interest in the population and given period. In the case of COVID-19, I show that for a given population the value of that weighted average only changes very slowly over time, providing an easy short-cut for fine-grained tracking of the impact of pandemic on the individual lifespan.

## Conceptual detour

Assessing PLEB reductions induced by a specific cause or event requires two period life tables, one representing the prevailing mortality conditions and another one representing the counterfactual mortality conditions expected in the absence of that cause or event. The assessment involves a relatively copious amount of life table manipulations, but decades ago Nathan Keyfitz provided most useful insights as to what these manipulations boil down to. Considering

the related issue of estimating the increase in PLEB brought by the permanent elimination of a cause of death, he summarized that the increase "depends on the average time that elapses before the persons rescued will die of some other cause" [10]. Conversely, the decrease induced by a new cause of death depends on the average time that would have elapsed before the persons who died from the new cause would have died from other causes.

This average time can be derived from the synthetic cohort approach modelled in the period life table where each death at age $a$ from a cause $C$, $d^C(a)$, reduces the number of person-years lived by the life expectancy at age $a$ in the absence of that new cause, $e^{o-C}(a)$. This commonly assumes that persons dying from the new cause would have had the same life expectancy in the absence of that cause as same-age persons who survived that cause. This common assumption may appear unlikely, and interactions between causes of death can be incorporated instead, but the data requirements are substantial. Under the common assumption, the difference in PLEB is thus the average over all members of the synthetic cohort, $l_0$ (the radix of the life table), of the difference in person-years lived by cohort members:

$$\Delta PLEB = \frac{1}{l_0} . \int_0^\omega (d^C(a).e^{o-C}(a))da \qquad (1)$$

Keyfitz' insight relates to the concept of "potential years of life lost" [11] developed a couple of decades earlier still. The initial approach, designed to measure premature mortality, compared ages at death to a fixed value (70 or 75 years) [12]. This approach is not suited to study cause of deaths at older ages since deaths at ages above the fixed value are not considered [13]. In burden-of-disease assessments, it has become customary to estimate Years of Life Lost (YLL) as:

$$YLL = \int_0^\omega (D^C(a).e^{o*}(a))da \qquad (2)$$

where $D^C(a)$ is the number of deaths from a certain cause $C$ at age $a$ observed in the population during a reference period and $e^{o*}(a)$ is life expectancy at age $a$ in a counterfactual life table. YLL to COVID-19 have been estimated using this approach [14–16].

Three differences between the two above equations can be observed. First, YLL are estimated from actual numbers of deaths by age rather than from numbers of life table decrements. This implies that the YLL estimate is sensitive to the age distribution of the population, as in turn it affects the distribution of deaths by age. While the PLEB is not properly speaking an age-standardized measure [17], it equals the inverse of the "stationary" death rate, that is, a weighted average of the period age-specific death rates with weights derived from these death rates through life table construction. Using these internally derived weights rather than an external, standard age distribution, the stationary death rate and PLEB are independent of the actual age composition of the population. This relative advantage of the difference in PLEB comes at the cost of using a "stationary" age distribution of deaths, however, represented by the life table decrements that result from indefinitely subjecting the population to the mortality conditions of the period. As discussed in the introduction, this assumption is precisely what is problematic for studying the impact of a mortality shock or an emerging disease such as COVID-19. Moreover, the actual distribution of deaths can be used in a population of any size and periods of any length, allowing for the mortality impact to be tracked on short temporal and small spatial scales for which interpreting differences in PLEB hardly makes sense.

The second difference refers to the counterfactual life expectancies. In global burden-of-disease assessments, a universal life table representing optimal survival conditions is typically used. This has the advantage of making YLL for different populations additive—allowing for the derivation of a global estimate of YLL due to a cause by simple summation. However,

using a universal life table may misrepresent the actual gains from averting a death in a specific population.

The last difference concerns the denominator, or lack thereof in YLL. Unnecessary for comparisons between different causes of death in the same population, for which YLL was initially intended, the introduction of a denominator is required for meaningful comparisons across populations [18]. At least three denominators for the YLL can be found in the literature. The first one is the total number of years that can be expected to be lived by members of the population, of which YLL is a fraction. That fraction is thus a ratio of death over population size, with both deaths and population members weighted by life expectancies at their respective ages. This yields a coherent measure with potential years lived in both the numerator and the denominator, but not one that can be interpreted in terms of individual longevity (in years per person). Another denominator is the total population size, which yields a ratio in years person. The corresponding ratio is a less coherent measure, however, as it includes in the denominator all the individuals in the population, including many that survived the mortality shock and do not contribute to YLL in the numerator. In turn, this complicates providing a precise interpretation for the value of YLL per capita. A third denominator is the number of deaths from the cause of interest in the population during the reference period. This yields an average YLL (AYLL) per such deaths, $D^C$:

$$AYLL = \frac{\int_0^\omega (D^C(a).e^{o*}(a))da}{D^C} = \int_0^\omega \left( \frac{D^C(a)}{\int_0^\omega D^C(a)da} \right).e^{o*}(a)da \tag{3}$$

The AYLL thus represents the average (universal) life expectancy of the population members who died from the specific cause during a given period. On the one hand, it is a coherent measure as its denominator now includes only the population members who died and contributed to lost years in the numerator. On the other hand, it is only a function of the distribution of deaths by age, irrespective of the prevalence of that cause of death. The AYLL thus cannot provide a measure of the intensity of a mortality shock.

Considering the advantages and limitations of the extant measures, I propose to add one measure of the mortality impact of a cause of death or an event on the individual lifespan. This measure, the Mean Unfulfilled Lifespan (MUL), is intended for situations where the underlying assumptions of PLEB might be implausible, and thus based, as the estimate of YLL, on actual numbers of deaths in a population during a period rather than on life table decrements. To retain its intuitive interpretation, however, the MUL is structured like the difference in PLEB as summarized by Keyfitz and, using counterfactual life expectancies representing the mortality conditions in the population of interest, similarly expressed as an average difference in person-years lived per person. Since the life table radix, $l_0$, equals the sum of all decrements at all ages, the structural equivalence is maintained by defining the MUL as:

$$MUL \overset{\text{def}}{=} \frac{1}{D}.\int_0^\omega (D^C(a).e^{o-C}(a))da \tag{4}$$

where $D$ is the total number of deaths (from all causes at all ages) during the reference period.

This intuitive interpretation of the MUL can be derived by rewriting this defining equation as:

$$MUL = \frac{\int_0^\omega ((D^C(a).(a + e^{o-C}(a) - a)) + (D^{-C}(a).(a - a)))da}{\int_0^\omega (D^C(a) + D^{-C}(a))da}$$

$$= \frac{\int_0^\omega ((D^C(a).(a + e^{o-C}(a))) + (D^{-C}(a).a))da}{\int_0^\omega (D^C(a) + D^{-C}(a))da} - \frac{\int_0^\omega ((D^C(a) + D^{-C}(a)).a)da}{\int_0^\omega (D^C(a) + D^{-C}(a))da} \tag{5}$$

where $D^{-C}(a)$ is the number of deaths from all causes but $C$ at age $a$ observed in the population during the reference period. The second term represents the average age at death in a given period. If we assume that individuals who die of other causes than $C$ die at the same age as they would have in the absence of cause $C$ (no indirect effect of cause $C$ on other causes of death), and that individuals who die of cause $C$ at age $a$ would have otherwise lived to age $a+e(a)$, the first term represents the average expected age at death of the same individuals in the absence of cause $C$. The difference between the average age at death in the population during a given period and the average expected age at death of the same individuals in the absence of cause $C$, the MUL can thus be interpreted as a measure of premature mortality for an average person dying in a population and during a period of interest.

The assumption that cause $C$ has no indirect effect on other causes of death is actually not required. As will be shown below, one can also derive the MUL from data on all deaths, and distinguishing between deaths that, based on counterfactual "benchmark" mortality conditions, were expected to occur in that period and those that were not, use the latter number of "excess" deaths instead of deaths from a specific cause. Finally, note that the MUL differs from changes in average ages at death across periods, which can be readily measured but may actually be positive with the emergence of a new cause of death if that cause affects people who are older on average that those dying from other causes. To sum up this conceptual detour, the MUL complements existing indicators of the impact on the individual lifespan of a cause of death by providing a measure of the average potential years of life lost to a specific cause of death, or to any mortality shock, for all population members dying in a certain period, regardless of their cause of death.

## Empirical shortcut

Calculating the MUL for small geographical areas and short periods is not conceptually problematic since it captures the actual length of lives that ended there and then, unlike differences in life expectancies whose interpretation in terms of reduced longevity requires assumptions about future conditions. The demand on data (including a separate counterfactual life table for each population of interest) is substantial, however, and the life table manipulations are not particularly straightforward.

To simplify the estimation, the MUL can be rewritten as:

$$MUL = \frac{D^C}{D} . \frac{\int_0^\omega D^C(a).e^{o-C}(a)da}{D^C} = \frac{D^C}{D}.PAYLL \tag{6}$$

The second term is similar to the AYLL in Eq (3), but again based on population-specific, counterfactual life expectancies instead of a universal one, and is termed here the Population AYLL (PAYLL) to underscore that difference. In a given population, the PAYLL is a weighted average of counterfactual life expectancies that are estimated from prior conditions. While these do not change over time., their weights are the ratios:

$$\frac{D^C(a)}{D^C} = \frac{M^C(a).N(a)}{M^C.N} = \frac{M^C(a)}{M^C} . \frac{N(a)}{N} \tag{7}$$

where $M^C$ and $M^C(a)$ are the all-age death rate and the death rate at age $a$ from a specific cause, and $N$ and $N(a)$ are the total population size and number of individuals at age $a$ in the population. These weights should also be expected to vary little within short periods because their values depend on the population composition, $N(a)/N$, and on the age pattern of cause-specific death rates, $M^C(a)/M^C$, both of which should vary little within short periods.

This suggests that the value of PAYLL can be expected to only change slowly over time and to be relatively close across populations with similar life expectancies and population

compositions. MUL values for a sub-population or during a sub-period can then be approximated as the product of the PAYLL for the whole population or the entire period and the time-varying all-age ratio of deaths from a specific cause to all deaths, $D^C/D$, in the sub-population or during that sub-period. As noted above, this can be readily extended to any mortality shock, with data on excess deaths and all-cause deaths.

## Materials and methods

### Methods

The equations defining YLL and the MUL appear deceivingly simple. To implement them, one has to apply estimates of life expectancies, which refer to exact ages, to numbers of deaths that are available or can be estimated only for age intervals. The varying value of life expectancy on a closed age interval is typically approximated by linear interpolation [19], in which case the contribution of the closed age interval between ages $x$ and $x+n$ to the MUL equals:

$$\frac{\int_0^n D^C(x+a).e^{o-C}(x+a)da}{D} = \frac{{}_nD_x^C}{D}.e^{o-C}(x+{}_na_x^C) \tag{8}$$

where ${}_nD_x{}^C$ is the number of individuals between ages $x$ and $x+n$ who died of cause $C$ during the reference period and ${}_na_x{}^C$ is the average number of years lived after age $x$ by these individuals. In turn, life expectancy at exact age $x+{}_na_x{}^C$ can be derived by linear interpolation between the values of life expectancy at ages $x$ and $x+n$.

The linear interpolation is more problematic for wider the age intervals and for older age groups. In the 2018 US life table for males for instance [20], life expectancy declines by 2.9 years between ages 75 and 80 and by 2.4 years between ages 80 and 85. In this case, the linear approximation over-estimate life expectancy in the interval. The average number of years lived after age 75 by individuals dying between ages 75 and 85 is 5.4 years, and linear interpolation would yield a life expectancy at age 80.4 years of 8.5 years, whereas life expectancy has already dropped to 8.4 years at age 80.

This upward bias is particularly undesirable if individuals dying of the cause(s) of interest might be expected to suffer from other long-term conditions that would increase their risk of mortality from other causes. For COVID-19 victims, for instance, a correctly estimated life expectancy for their exact age would already over-estimate their potential years of life lost by ignoring well-documented "co-morbidities". As for the open-ended interval, linear approximation requires setting a somewhat arbitrary upper age limit. Relatively minor for premature mortality at relatively young ages, these issues make linear approximation more problematic to apply when the cause(s) of interest affect older individuals. In the case of COVID-19 deaths for instance, close to 60% of these deaths were above age 75 years (Fig 1) and reported in just one ten-year closed interval (75 to 84 years) and one open age interval (over 85 years).

In this respect, working with all deaths in a given period rather than with deaths from a specific cause of interest presents an important empirical advantage in addition to the benefit of assessing both the direct and the indirect effects of that cause of death. Using all deaths in a closed age interval between ages $x$ and $x+n$, ${}_nD_x$, one may first calculate the contribution to the MUL of the deaths that were expected to occur under the counterfactual mortality conditions, ${}_nD_x{}^{-C}$. For these deaths, the difference in length of life averages to the difference between the average number of years lived in the age interval under the prevailing conditions, ${}_na_x$, and under the counterfactual conditions, ${}_na_x{}^{-C}$. The advantage of this approach is that setting an arbitrary upper value for the open-ended age interval becomes unnecessary. Having already reached age $N$, all the individuals who died in the open age interval in the period would all have been expected to die in the same open-ended age interval under the counterfactual

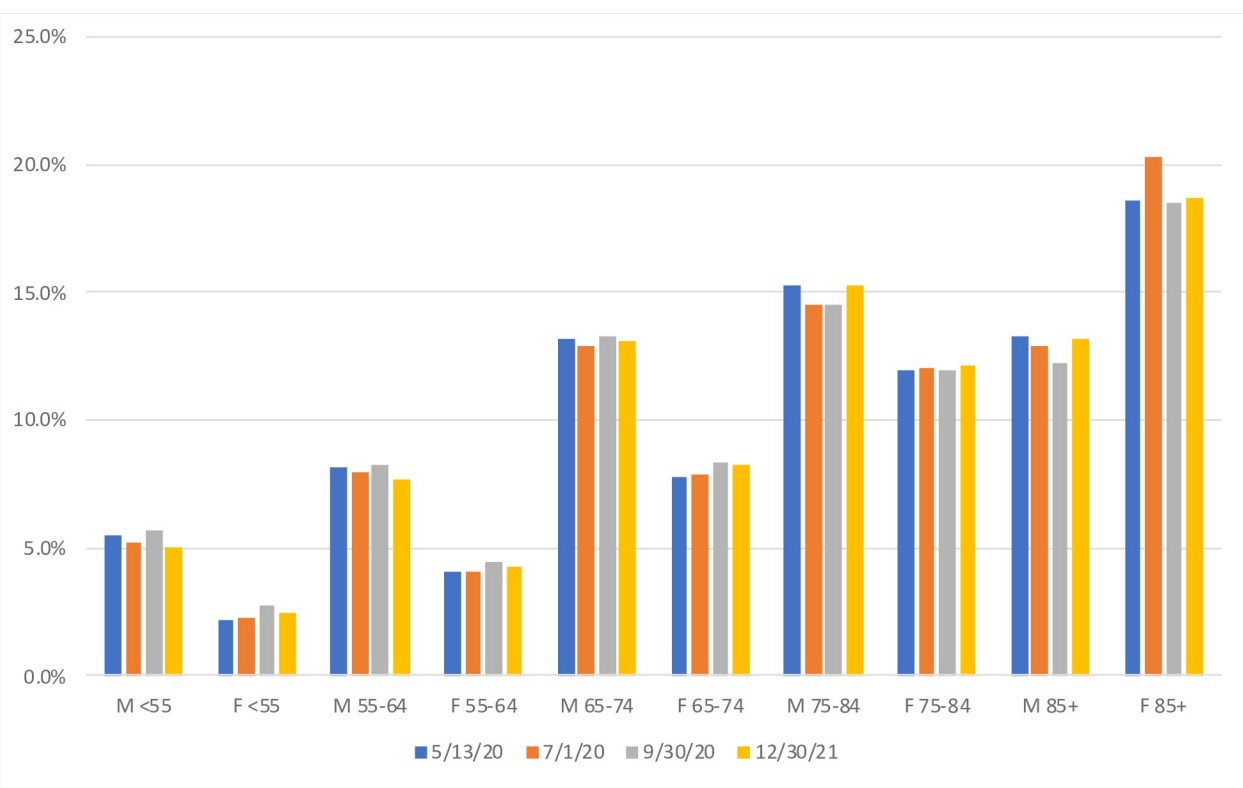

**Fig 1. Distribution of provisional COVID-19 death counts, by sex- and age-groups, USA as of 5/13 (54,860 deaths), 7/1 (112,223 deaths), 9/30 (194,087 deaths) and 12/30 (301,671 deaths).** Source: Centers for Disease Control and Prevention (CDC).

mortality conditions, albeit not necessarily in that same period. For all the open-interval deaths over age $N$, $D_{N+}$, the difference in length of life thus averages to the difference between life expectancy at age $N$ under the prevailing conditions, $e^o(N)$, and under the counterfactual conditions, $e^{o-C}(N)$. With this approach, only the issue of estimating the average reduction in length of lived lives on closed age intervals from values of life expectancies that vary on these age intervals remains, and pertains to "excess" deaths, ${}_nD_x - {}_nD_x^{-C}$.

An alternative to linear interpolation derives from the fact that an individual's expected length of life ($e^o(x)+x$) gradually increases with age $x$. Life expectancy at any age is thus larger than the difference between life expectancy at an earlier age and the difference between the two ages:

$$\int_0^n D(x+a).e^o(x+a)da > \int_0^n D(x+a).(e^o(x)-a)da = {}_nD_x.(e^o(x) - {}_na_x) \qquad (9)$$

Applying this approximation to excess deaths on any closed age interval will induce some underestimation of the average life expectancy of individuals dying on the interval. As discussed above, however, this may be preferable to the overestimation induced by linear interpolation in situations where ignoring co-morbidities is likely to already entail some overestimation of potential years of life lost.

Adding the contributions of the different types of deaths yields:

$$MUL \cong \frac{1}{D} \cdot \left( \sum_{x=0,n}^{N-n} {}_nD_x^{-C}.({}_na_x^{-C} - {}_na_x) + (D_{N+}.(e^{o-C}(N) - e^o(N))) + \sum_{x=0,n}^{N-n} ({}_nD_x - {}_nD_x^{-c}).(e_x^{o-c} - {}_na_x) \right) (10)$$

Rearranging the sums corresponding to the closed age intervals and using the multiple-decrement life table relationship $e^o(N)/e^{o-C}(N) = D^{-C}_{N+}/D_{N+}$, this can be rewritten as:

$$MUL \cong \frac{1}{D}\left(\sum_{x=0,n}^{N-n}({}_nD_x \cdot ({}_na_x^{-C} - {}_na_x) + ({}_nD_x - {}_nD_x^{-C}) \cdot (e_x^{o-C} - {}_na_x^{-C})) + ((D_{N+} - D_{N+}^{-C}) \cdot e_N^{o-C})\right) (11)$$

## Materials

To illustrate the alternative approximation in Eq (11), I estimate quarterly MUL in the last three quarters of 2020 and the first quarter of 2021 in different national and sub-national populations. This requires for each of these populations life table values of $e_x^{o-C}$ and ${}_na_x^{-C}$ corresponding to survival conditions in the absence of COVID-19 whose. Combined with the number of individuals by sex and age-group, the life table values of age-specific death rates, ${}_nm_x^{-C}$, then provide the expected numbers of deaths ${}_nD_x^{-C}$ in the absence of COVID-19. National population by age and sex as of July 1, 2020 and life table functions for 2015–20 and 2020–25 were obtained from the UN Population Division [21]. From these, linear and exponential interpolation yielded counterfactual ${}_nm_x^{-C}$ and ${}_np_x^{-C}$ values, from which $e_x^{o-C}$ and ${}_na_x^{-C}$ were then derived using life table relationships. Population data and life tables for sub-national populations in Italy, Mexico, Peru and the US were obtained from national statistical agencies [22–27].

New population exposure and life tables representing actual mortality conditions (with COVID-19) were derived for each quarter to provide the corresponding values of ${}_na_x$. The construction of these life tables requires quarterly numbers of deaths by sex and age-group, ${}_nD_x$ [28]. In countries where vital statistics are incomplete not available yet, but estimates of COVID-19 deaths are available, these numbers, ${}_nD_x$, can be obtained by adding estimates of COVID-19 deaths to the counterfactual number of deaths in the absence of COVID-19, ${}_nD_x^{-C}$, through a multi-decrement life table to adjust for competing risks of deaths. When estimates of COVID-19 deaths are not broken down by sex and age-group, an alternative is to use a reference set of age-and-sex death rates from COVID-19 from another population for which these rates are deemed reliable [29]. Centers for Disease Control and Prevention (CDC) data for the USA in 2020, the country with the largest number of COVID-19 deaths in 2020, were used here as the reference set of age-and-sex death rates from COVID-19 [30]. Data on COVID-19 deaths by sex and age-group from Brazil [31] and the Netherlands [32] were used to investigate the results sensitivity to this assumption and to the CDC's age groupings.

Quarterly MUL and PAYLL were estimated from number of COVID-19 deaths by March 31, June 30, September 30 and December 31, 2020, and March 31, 2021 for national and sub-national populations provided by the John Hopkins' Coronavirus Resource Center [33]. Daily values from the same source were also used to illustrate the short-cut in Eq (6). Another application of Eq (6) for small-area populations used data on the monthly number of deaths by province in Ecuador [34].

All of these data are publicly available and were downloaded from institutional websites. Full results and sensitivity analyses are provided in Excel spreadsheets in the Supplementary Files. R routines and intermediate data files (e.g., counterfactual life tables) are available on a Github repository (heuveline/ind-comp-mort).

## Results

Fig 2 shows quarterly MUL for both sexes combined in four selected sub-national populations: Lombardy (Italian Region), New Jersey (US State), Mexico City (Mexican State), and Lima (Peruvian Department). These populations were selected to illustrate trends in MUL because

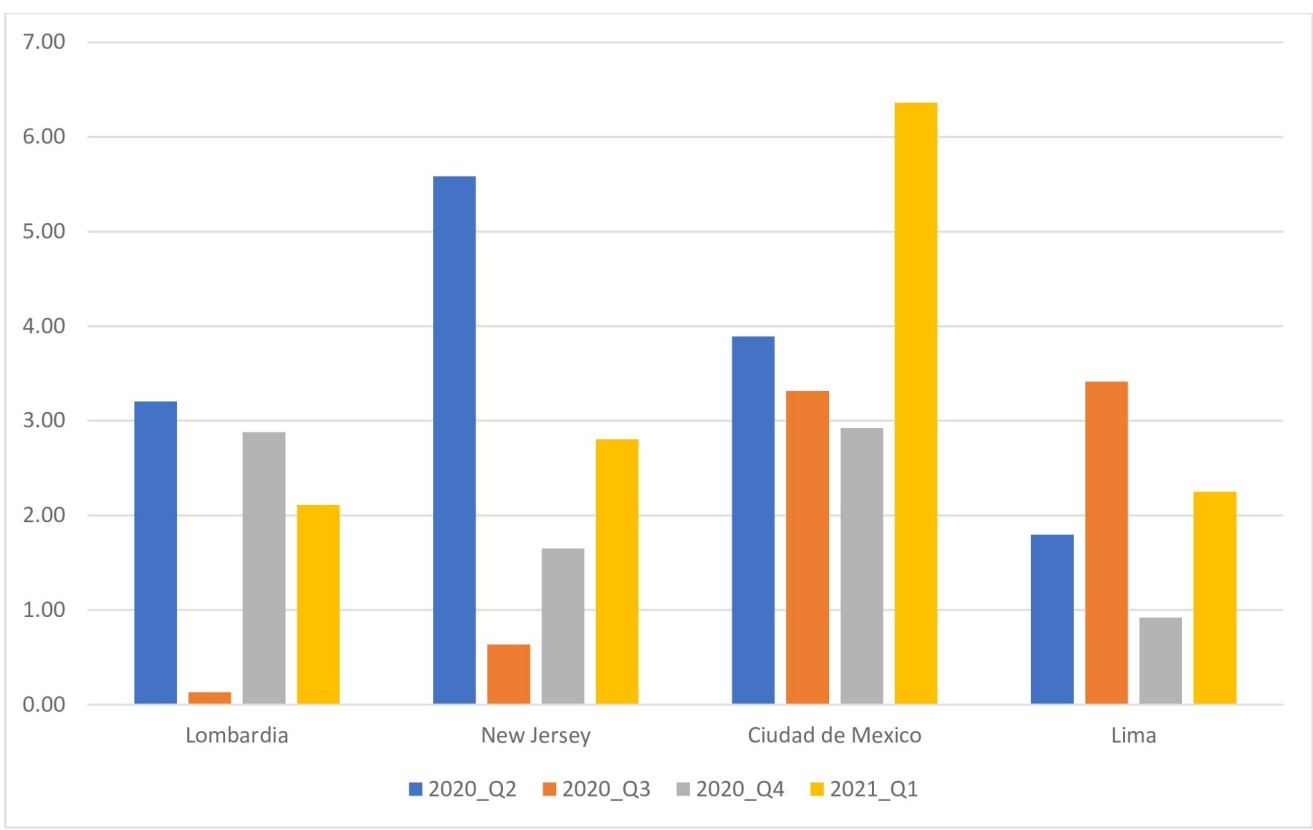

**Fig 2. Quarterly Mean Unfulfilled Lifespan (MUL) for both sexes, in years, selected populations.**

they have comparable population sizes (in the of order of 10 million) and belong to regions of the world that the pandemic reached at different times. Quarterly MUL illustrate the peaking of a first wave of COVID-19 deaths in Lombardy, New Jersey and Mexico during the second quarter of 2020. While in the third and fourth quarter a decline and rebound are clear in Lombardy and New Jersey, the trend was comparatively flat in Mexico City. A second wave peaked in the last quarter of 2020 in Lombardy, but continued to develop into the first quarter of 2021 in New Jersey and Mexico City. As measured by MUL, the two waves had comparable impact on the individual lifespan in Lombardy, but the second wave was less impactful than the first in New Jersey whereas it was the opposite in Mexico City. In Lima, the trend appears to be the reverse of that of Lombardy. The first wave peaked in the third quarter of 2020 and the second wave began in the first quarter of 2021.

For a finer-grained representation of the temporal trend within a time period, Eq (6) can be used with the assumption that PAYLL are population specific but invariant over that period. The estimated quarterly PAYLL (see S1 Table) confirm the variations across populations, from a low of 9.6 years in Bulgaria to a high of 26.1 years in Qatar. These differences can be explained by different age compositions, with younger compositions giving more weight to remaining life expectancies at younger ages, which are higher. As expected, however, in any given population, PAYLL changes relatively little from one quarter to the next. Building on this, Fig 3 shows MUL for a rolling seven-day period, derived from daily numbers of COVID-19 deaths and second-quarter PAYLL in Lombardy, New Jersey and Mexico City, from mid-March to mid-July 2020. First-wave peak MUL were reached at the end of March in Lombardy (6.43 years), three weeks later in New Jersey (8.91 years) and not until early June in Mexico City (6.24 years).

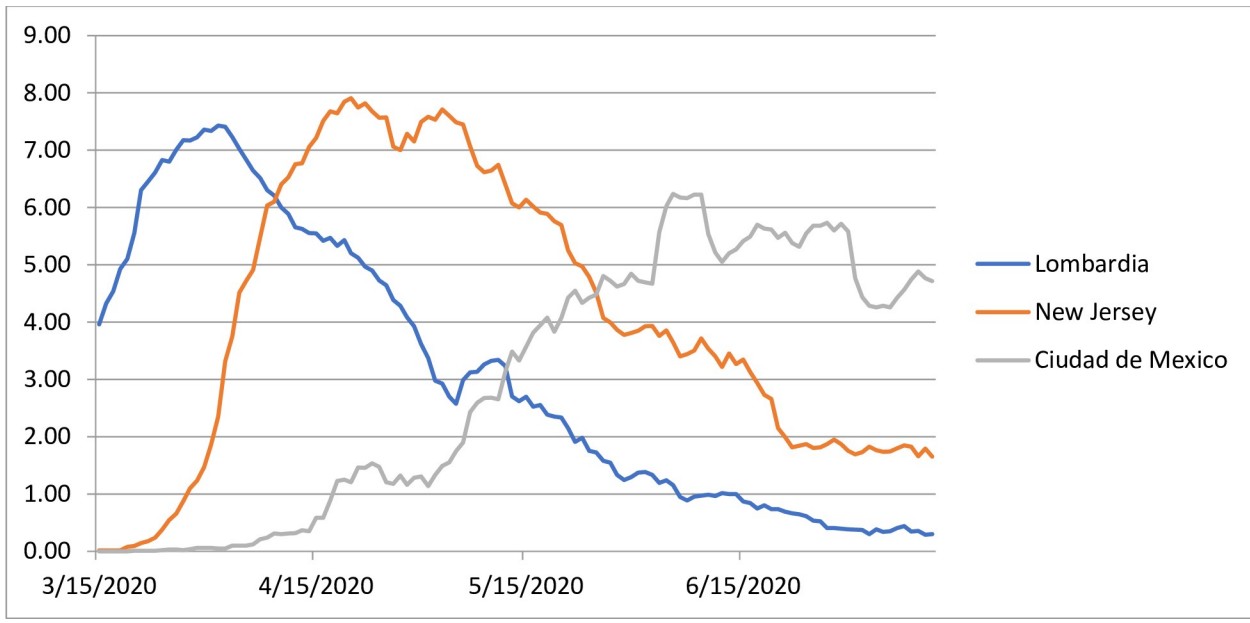

**Fig 3. Seven-day Mean Unfulfilled Lifespan (MUL) for both sexes, in years, selected populations.**

Eq (6) can also be used to approximate MUL for small populations for which life table functions might not be available, by "borrowing" the PAYLL from a larger population. This can be illustrated for the province of Guayas, Ecuador, where the monthly numbers of deaths show a marked increase in March, April and May. From a baseline of 1,700–2,000 per month in January, February and again in June, the number of deaths in April reached 12,425. This indicates that 85.2% of deaths in that month might be considered "excess" death. Based on this ratio and the second-quarter PAYLL derived for Ecuador (14.9 years, S1 Table), Eq (6) suggests that individuals died 12.7 years younger, on average, that their expected age at death in April in the province of Guayas. Illustrating the intensity of the mortality shock in Guayas, this monthly MUL is substantially larger than even the peak seven-day MUL shown in Fig 3 for Lombardy, New Jersey and Mexico City. It should be noted, however, that the Guayas estimate was derived from excess deaths, whereas those in Fig 3 were derived from COVID-19 deaths only —a difference discussed in the next section.

## Discussion

The MUL is proposed here as an alternative to induced changes in PLEB to assess the impact of a cause of death on the individual lifespan for situations where the assumptions underlying life table construction are implausible, invalidating the usual interpretation of the PLEB. Complementing existing measures based on YLL, MUL is similarly based on estimating the number of potential years of life lost corresponding to deaths from a specific cause or excess deaths from a specific event in a given period. As directly comparing Eqs (1) and (4) shows, by averaging estimated potential years of life lost over the total number of deaths in the period, MUL is structured like a difference in PLEB. The MUL retains the intuitive interpretation of a difference in PLEB as a change in average length of life, but for an actual cohort of individuals (those dying in the period) subjected to prevailing mortality conditions rather than for a synthetic cohort subjected to indefinitely constant conditions.

To illustrate the derivation and interpretation of the MUL in the context of COVID-19 mortality, quarterly and seven-day MUL values were derived for different populations. As argued above, the MUL is best calculated from estimates of excess deaths by age and sex derived from period data on all-cause mortality by age and sex. In the case of COVID-19, a number of countries with good vital statistics were affected early on and considerable effort went into producing excess mortality estimates for these countries in a remarkably short time [35]. In the majority of the affected countries, however, the absence of reliable civil registration statistics precludes the indirect estimation of excess mortality—a limitation that can be expected to be relatively typical on other instances of mortality shocks. Moreover, even in countries with good vital statistics, data might not be available at the sub-national levels. To illustrate MUL derivation with these data limitations, quarterly MUL values were calculated instead from quarterly numbers of total (all-age) COVID-19 deaths in the population. This involved four assumptions: (1) COVID-19 deaths are properly accounted for in each population (unbiased "total" estimate), (2) all populations share an age-and-sex COVID-19 mortality "pattern" (i.e., the same ratio of a given age-and-sex-specific death rate to the all-age, both-sex death rate), (3) death rates from other causes are unchanged (the number of deaths from other causes is only reduced to the extent that COVID-19 deaths reduce exposure to other causes of deaths), and (4) individuals who died from COVID-19 would have faced the same risk of death from other causes as any individual of the same age and sex.

As discussed above, the last one is a standard assumption of YLL-based measures, but nonetheless problematic in the case of COVID-19 due to the higher proportion of several underlying long-term conditions (e.g., obesity) observed among COVID-19 victims. Adjusting for differences in long-term conditions prevalence is very data demanding, however, and the impact might not be as large as expected. A study performing this adjustment found that it reduced the AYLL from COVID-19 in the United Kingdom from 13 to 12 years (average for both sex) [36]. An alternative strategy was proposed here that underestimates the value of potential years of life lost over an age interval by estimating its value at the beginning of the age interval. As shown in the supplementary files, this yields quarterly PAYLL values for the United Kingdom that vary between 11.9 and 12.0 years (S1 Table), values thus quite close to what the adjustment for underlying long-term conditions might have provided. While there is of course no guarantee that this strategy would apply equally well to causes of death other than COVID-19, ignoring comorbidities will likely lead to some overestimation of AYLL for most causes of death, and purposely underestimating the number of potential years of life lost will likely be preferable in other situations as well.

With respect to the second assumption, age patterns of COVID-19 death rates are becoming available for an increasing number of nations [37], and do appear to share exhibit strong regularities with only modest variation in their slope except at the oldest ages [38]. The latter seems to reflect higher infection rates in nursing homes in some European countries and the USA [39]. There is also evidence of limited shifts in the age and sex patterns over time that might be related to policies implemented to mitigate fatality rates among the most vulnerable [40]. As shown in Fig 1 comparing the distribution of provisional COVID-19 death counts in the USA at four points in 2020 (May 13, one of the earliest dates for which this distribution is available, July 1st, September 30th and December 30th), death rates in the oldest age groups might have declined relative to the death rates at other ages, but the changes were quite modest.

Two sensitivity analyses were conducted to assess the effects of potential deviations from the second assumption (see Supplementary Files). The first one substituted the actual sex and age pattern in Brazil, the country with the second largest number of COVID-19 deaths in 2020, to the one derived by assuming the same COVID-19 mortality pattern as in the USA. As

the slope of the Brazilian age pattern is a little less steep its US counterpart, the substitution makes the distribution of the COVID-19 deaths younger, increasing quarterly MUL estimates by about 8 to 9% (e.g., from 2.75 to 2.92 years at its peak, which in Brazil is reached in the third quarter of 2020, as shown in Fig 2 for Peru). This suggests that using population-specific data on COVID-19 mortality by age and sex might would improve MUL estimates, but not drastically so.

The second sensitivity analyses focused on the limitation of having data categorized as by the CDC, in 10-year closed age groups and for an oldest age group of 85 years and over. This open age-group may include a substantial share of COVID-19 victims (see Fig 1 for the USA), with substantial differences in YLL within this group. This was investigated with data for the Netherlands, as the country provides data for 5-year closed age groups and the open age group 95 years and over, estimating quarterly MUL with this age breakdown and after regrouping deaths as categorized by CDC. MUL values estimated with the original breakdown were about 8% smaller than with CDC age intervals, with the open-ended interval accounting for a little less than half of the difference. As shown for Lombardy in Fig 2, quarterly MUL in the Netherlands were larger in the second and fourth quarter, reaching .95-.96 years. With the open age interval starting at 85 years, the quarterly estimates increase to .98 years, and additionally, with 10-year closed age intervals, to 1.04 years. The overestimation induced by coarse age grouping is consistent with the fact that the age-pattern of COVID-19 mortality is a little steeper than the age-pattern of overall mortality [41] but again the overestimation is relatively modest.

In the case of COVID-19, the main concerns with respect to MUL estimation relates to the first assumption, namely, the proper reporting of COVID-19 deaths (first assumption). The potential biases discussed above with respect to the second and fourth assumption amounted to less than 10% changes in one direction or the other, but the ratios of excess deaths to reported COVID-19 deaths estimated for some countries (e.g., 2.2 in Mexico or 2.7 in Peru) [42] indicate that less than half of their COVID-19 deaths might have been accounted. If the discrepancy between excess mortality estimates and reported COVID-19 mortality correspond to under-reported COVID-19 deaths, Eq (6) can be used with an adjusted ratio $D^C/D$. (If $k$ is the ratio $D^C/D$ derived from reported COVID-19 deaths and $\alpha$ is the ratio of excess to reported COVID-19 deaths, the initial value of $k$ should be multiplied by $\alpha.k/(1+(k.(\alpha-1))$ to obtain the ratio of excess to total deaths and, based on Eq (6), so should the initial MUL value). With ratios of 2.2 in Mexico and 2.7 in Peru for instance, the peak 7-day MUL values would increase from 6.24 to 8.96 years for Mexico City and from 2.67 to 5.49 years for Lima.

Adjusting MUL values in this manner should perform well when excess deaths do not also include excess deaths from other causes. Data from countries with good vital statistics, however, provides evidence against the assumption that death rates from other causes did not change (third assumption). In a number of high-income nations, declines in death rates from other causes are suggested by fewer excess deaths than reported COVID-19 deaths alone [43], whereas US death rates from heart disease and unintentional injuries have increased markedly during the pandemic [44]. In situations where a substantial portion of the discrepancy between estimates of excess deaths and official numbers of COVID-19 deaths originates in changes in other causes of deaths, the above adjustment of the MUL value through Eq (6) will not perform as well. In the case of other death rates that increased during the COVID-19 pandemic, such as unintentional injuries, because they tend to be at younger ages than COVID-19 deaths, equating excess deaths to unreported COVID-19 would lead to some underestimation of years of lost life and, as a result, MUL. A similar gap can be observed considering estimates of changes in PLEB that are now available in a number of nations [45] and that differ more from initial estimates based on reported COVID-19 deaths only [46] than excess to COVID-19 death ratio would have suggested.

Overall, MUL is not only structurally similar to a difference in PLEB, but its computation is also similar to that of a change in PLEB induced by incorporating additional deaths in a multiple-decrement life table framework. Similar assumptions are required, and both estimates are likely sensitive to a similar degree to the data quality issues discussed above, from the total number of additional deaths to their age distribution. The main difference between the two indicators is that, like YLL-based measures, MUL is not sex- or age-standardized and MUL comparisons across populations that differ markedly in age composition will be biased. On the one hand, all else equal, a younger population composition yields a younger distribution of deaths and a higher PAYLL value. On the other hand, when the cause of interest is one that affects older individuals more than all-cause mortality, as is the case for COVID-19, an older population distribution contributes to a higher proportion of excess deaths relative to all deaths. The two, opposite age-composition effects may partially off-set each other, but unlike differences in PLEB, MUL remains dependent on population distribution. This advantage of differences in PLEB over MUL comes at a substantial cost, however, since the internal derivation that rids differences in PLEB of the influence of population distribution assumes that period mortality conditions will become permanent. Almost by definition, this assumption is not tenable in the case of a mortality "shock."

In such situations, MUL provides an unstandardized alternative to differences in PLEB, more readily interpretable as an average difference in length of lives lived per person. Related to other unstandardized measures such as the AYLL or YLL per capita, the MUL's interpretation pertains to an actual death cohort, that is, population members who died during a certain period rather than to a synthetic cohort as represented in the life table. To reiterate, the MUL indicates the difference between their average age at death and their average expected age at death had a temporary mortality shock not occurred. This interpretation does not build on any assumption that these temporary conditions will either pass entirely or extend indefinitely into the future. Moreover, MUL remains interpretable regardless of its temporal and geographical scale, pertaining to deaths in a given population and period, and can serve as a short-term, micro-level indicator. As long as PAYLL can be assumed to be almost constant on a period, MUL values can easily be approximated as the product of the corresponding PAYLL and the ratio of excess to total deaths in the population during the period. In the specific case of COVID-19 mortality, quarterly values were found indeed to change little from one quarter to the next (S1 Table). This approximation can also be used for sub-area populations for which the necessary data are only provided of the whole-area population, as long as the age compositions of the sub- and whole-area populations can be held as relatively similar.

## Supporting information

**S1 Table. Full results.** MUL and related indicators for all national and first-level sub-national populations in Brazil, China, Italy, Mexico, Peru, Spain, and the United States of America.
(XLSX)

**S1 File. Shortcuts.** Estimation of MUL values for sub-periods or sub-populations in Lombardy, New Jersey and Mexico City (Fig 3) and in Guayas, Ecuador.
(XLSX)

**S2 File. Sensitivity Brazil.** Sensitivity analysis using actual age distribution for Brazil.
(XLSX)

**S3 File. Sensitivity Netherlands.** Sensitivity analysis using actual age distribution for the Netherlands.
(XLSX)

## Acknowledgments

The author thanks Hiram Beltrán-Sánchez for life tables for Mexican States, and Michel Guillot, Philippe Bocquier, Tim Riffe, Michel Garenne, Christophe Guilmoto and Sam Preston for comments on an earlier draft of this manuscript.

## Author Contributions

**Conceptualization:** Patrick Heuveline.

**Formal analysis:** Patrick Heuveline.

**Investigation:** Patrick Heuveline.

**Methodology:** Patrick Heuveline.

**Validation:** Patrick Heuveline.

**Visualization:** Patrick Heuveline.

**Writing – original draft:** Patrick Heuveline.

**Writing – review & editing:** Patrick Heuveline.

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
