## [Decision Letter · Decision Letter 0]

26 Mar 2021

PONE-D-21-05770

The Mean Unfulfilled Lifespan (MUL): A new indicator of the impact of mortality shocks on the individual lifespan, with application to global 2020 quarterly mortality from COVID-19

PLOS ONE

Dear Dr. Heuveline,

Thank you for submitting your manuscript to PLOS ONE. After careful consideration, we feel that it has merit but does not fully meet PLOS ONE’s publication criteria as it currently stands. Therefore, we invite you to submit a revised version of the manuscript that addresses the points raised during the review process.

We, reviewers and myself, that the paper makes an extremely important contribution to the discussion. The main idea of the paper is very innovative. However, we believe the paper needs some adjustments before it can be published. In general, the paper could be better structured, figures should be revised for clarity, author could make replicable (I believe other will be interested in applying the method), and there are some issues in presenting the material that could be improved. For instance, I would like to see a better explanation of the calculations with more detailed discussion of examples.  The reviewers made very valuable and detailed comments. 

Some suggestions/comments

selection of countries and examples. One suggestion is to focus on a series of countries with better data quality to be able to show in more detail calculations and discussion;this is related to the presentation of results and discussion. This section could be revised aiming for a more organized and structured presentation. I many parts the reader has some trouble following the main ideawould it be possible to perform sensitive analysis?improve presentantion of results - there are many issues with the figures that were also reported by reviewersimprove the discussion on the limitations of dataplease, see detail comments by both reviewers. 

We look forward to receiving your revised manuscript.

Kind regards,

Bernardo Lanza Queiroz, Ph.D

Academic Editor

PLOS ONE

Journal Requirements:

'No. The funders had no role in study design, data collection and analysis, decision to publish, or preparation of the manuscript.'

6. Please include captions for your Supporting Information files at the end of your manuscript, and update any in-text citations to match accordingly. Please see our Supporting Information guidelines for more information: http://journals.plos.org/plosone/s/supporting-information

Reviewers' comments:

Reviewer's Responses to Questions

**Comments to the Author**

1. Is the manuscript technically sound, and do the data support the conclusions?

Reviewer #1: Yes

Reviewer #2: Partly

2. Has the statistical analysis been performed appropriately and rigorously? 

Reviewer #1: Yes

Reviewer #2: I Don't Know

3. Have the authors made all data underlying the findings in their manuscript fully available?

Reviewer #1: No

Reviewer #2: No

4. Is the manuscript presented in an intelligible fashion and written in standard English?

Reviewer #1: Yes

Reviewer #2: Yes

5. Review Comments to the Author

Reviewer #1: The author proposes a new indicator of average mortality to reflect life lost during the pandemic. I find the article well written and innovative. Below find my suggestions and questions that could be clarified aimed at improving the paper.

1)There are many data quality and reporting issues across different dimensions (age, sex, timely data, all cause vs COVID 10, etc) for many of the populations that the author analyses. In addition, the author mentions an analysis of 159 national populations and 122 sub-national populations, but the results fail to show most of these and instead only show results for few populations. In my view, the main contribution of the article is the new indicator and find the sparse results distractive. Why not focusing on a set of countries/population with high quality data at the granularity that is required to conduct the analysis and provide a more in-depth description of these results? The author relies on many estimates without showing any sensitivity of the estimates or validation of the choices (for example to disentangle by sex deaths), when at the moment there is reported information that could be used to estimate MUL, for example leveraging the STMF from Human Mortality Database for some countries and the COVerAGE database project (Riffe et al 2021).

2)The expression ‘The MUL also equals the product of (1) the proportion of deaths in the population during a reference period that are due to a specific cause or event and (2) the average reduction in length of life among individuals who died from that specific cause or due to that specific event in the population during the reference period. In the case of COVID-19’ is confusing to me because the term ^{−} () is not a reduction but a counterfactual value of life expectancy in the absence of C. For me the ‘average reduction’ would be ^{} - ^{−} or something similar. I noticed a couple of sentences with this issue, could the author either rephrase this or am I understanding it wrong?

3)Another counterfactual calculation of life expectancy in the absence of a cause of death that is worth mentioning is that described in Beltran-Sanchez et al (2008) in Demographic Research. Is the one used by the author in this article analogous to the one described in the cause-deleted section in his co-authored book Preston et al?

4)How is PAYLL calculated? It is mentioned that is a weighted average of counterfactual life expectancies that are estimated from prior conditions and do not change over time, but then in the methods section it says that they come from statistical offices, could you elaborate more on this?

5)A sensitivity analysis on how results differ from 5-year age groups and single year age groups would be useful. Similarly, some discussion on open age interval and how sensible results are to it would inform the reader about limitations.

6)Another approach to the interpolation would be to ungroup cause-specific deaths into single age groups, which would potentially ease the assumption made over n_a_x. This is what is done in the COVerAGE database project (Riffe et al 2021) using a penalize composite link model proposed by Rizzi et al and implemented in the ‘Ungroup’ package in R.

7)When calculating the lifetables representing actual mortality conditions (with COVID-19) in each quarter, how did you deal with the exposures? For yearly lifetables we tend to use the mid-year population but for quarterly life tables these should be adjusted.

8)The graphs could be improved by labelling correctly each element.

9)The lack of reproducibility has been noted as an issue in the social sciences. I encourage the author to share data and code, or at least code, used to make the calculations in an open repository. This would not only help in the reviewing process but also encourage readers to use the indicator proposed.

Reviewer #2: The paper aims to fill the gap between conclusions drawn from the imaginary period life table population and the vital counts happening in the real populations. It is a deep and thoughtful work that constitutes a valuable and perhaps even pathbreaking contribution to demographic literature. It is fascinating to see that the challenges of the mortality shock in the pandemic year became the catalyst that pushed demographic thinking closer to answering the burning questions. I’m strongly convinced that the paper should be published. Yet, in the current form the substantial idea does not seem to be comprehensively and conclusively supported by empirical examples. Below follow my comments and suggestions that can hopefully help the author improve this impressive paper.

The main challenge of the paper is the balance between the initial idea-driven sound methodological contribution and the timely and demanding context that the c19 pandemic pose. Honestly, I think the author did not succeed much in balancing the two would be focal topics of the paper. Upon a careful read, I’m convinced that the paper is primarily about the method and not the c19 mortality shock. If so, it would be nice to use some well documented mortality shock, for example the heat wave of 2003 in Europe, to illustrate the efficiency of the method on solid and conclusively recorded data. This would also allow to illustrate the magnitude of the main limitations and enrich the sketchily outlined discussion of the assumptions with careful sensitivity analysis. It would also allow to mimic the data challenges that arise from the coarseness of the provisional data on c19 deaths – 10y age groups and the open-ended age interval – and get some idea of how much they affect the estimates. If in contrast the author is convinced that the main focus of the paper would be to provide a comprehensive comparative framework for overviewing the impact of c19 pandemic on mortality across the world, the empirical evidence and the way the results are presented should be radically improved and systematized, and many more data biases and processing challenges should be addressed in much finer details. It seems to me that this paper naturally separates in two outlined parts, of which the first seems to be much more developed in the current manuscript.

Results section is very difficult to follow since there is no apparent structure in the way results are presented, and the mentioned in abstract “159 national populations and 122 sub-national populations” are not readily presented. It seems, the author needs to choose if the paper is primarily methodological or empirical that addresses the hot topic of 2020 mortality shock. If the latter, a much clearer way to deliver and represent comparable results for the said populations and sub-populations should be designed as dumping those in Table S1 is simple barely digestible for the readers.

Looking at Fig 3 and the empirical shortcut formula, it seems MUL is limited from the top by PAYLL in the population based on the counterfactual life table. With a sharp spike of deaths, as was seen in Madrid and New York, and the granular time scale, it seems the assumed relevance of the counterfactual life table may be compromised. For example the paper Pifarré i Arolas etal 2021 (#15 in the ref list) clearly shows that the naïve matching of c19 deaths with a reference life table results in a much larger than usual estimate of the average YLL per c19 death. Fig 1 does not provide a required confirmation that the reference life table may be applied here. First, the estimates shown in Fig 1 do not account for the pre-pandemic age distribution of deaths. Second, the example considers the US population, in which the unfolding of the pandemic, while varying largely across states, went pretty uniformly at the federal level of aggregation – so the comparison of the quarter estimates within the pandemic year does not provide the support for the appropriateness of using a 2018 life table as a reference.

The author argues several times that the shorthand estimation of MUL relies on the stability of PAYLL. If one takes this for granted, PAYLL can be disregarded, in which case the measure reduces to a p-score – % increase in all-cause mortality over the baseline, which is routinely reported for c19 deaths – via a straightforward logit transformation [p-score = exp(logit(Dc/D))]. So, basically under the assumptions of MUL, the paper suggest a translation of p-scores into a YLL measure. This may be more intuitive but inherits the population age structure driven limitations of p-scores.

Discussion (and also the Introduction) seem to accept the widespread misinterpretation of PLEB as an indicator that (under the hefty assumptions that are mostly omitted in popular interpretations) can inform on the mortality shock effects at the individual level. PLEB is not designed to do so. In a way stating the advantage of MUL (specifically designed to answer this question) against PLEB is not fair. The very first sentence of the Abstract (line 22) formulates this intuitive misinterpretation of PLEB forming the basis of the unfair comparison.

What seems a bit confusing in the results is that by MUL the author means only the calculation for one cause of death, namely c19. Since MUL is generic and applicable to any cause of death in any context it would be nice not to build an association between MUL and c19 necessarily.

Much more detailed discussion of c19 data limitations and biases is required. For example, on Line 271 ”but estimates of COVID-19 deaths are available” – under- and untimely reporting happens even in the more developed countries. Karlinsky & Kobak (https://doi.org/10.1101/2021.01.27.21250604) provide an overview of how largely c19 deaths are underreported compared with excess deaths. Similarly, there is no discussion of the limitations imposed by the use of WPP reference life tables that are based on 5x5 Lexis grid and extrapolations. How exactly were the reference life tables constructed? Were the average 2015–19 WPP life table assumed representing the 2020 counterfactual good enough? A full replication package documenting all the data processing and calculation steps would help a lot.

Technical comments

Numbering the equations would improve the readability, cross-referencing, and (in future) citation of the paper.

Figures can be improved a lot. Axis titles are missing. All figures will benefit from optimizing the size of the text element – currently too small. Figures 1 and (especially) 2 can benefit a lot from flipping the axes, which would allow to place text horizontally – comfortable for the reader. Figure 1 may be aligned in a population pyramid like fashion. X-axis of figure 3 may be optimized to show some key ticks/dates, e.g. months.

Some preprints in the references have already been published in peer-reviewed journals, e.g. #15 is now doi.org/ 10.1038/s41598-021-83040-3.

Minor comments

Line 138: “left” is a bit confusing word in thee context, since these are years lost

Line 227: “because COVID-19 victims are more likely to suffer from other long-term conditions” – a reference is needed, possibly (https://www.medrxiv.org/content/10.1101/2020.10.19.20214494v2)

Line 253: “appears preferable in this case” – reasoning needed

Line 359: “CoViD-19” – written differently

Line 366: “improve MUL estimates by a similarly modest order of magnitude” – I may be mistaken as a non-native English speaker, but I’m used to think that “order of magnitude” refers to a roughly x10 magnitude, which is out of place in the context of the phrase.

6. PLOS authors have the option to publish the peer review history of their article (what does this mean?). If published, this will include your full peer review and any attached files.

Reviewer #1: No

Reviewer #2: **Yes: **Ilya Kashnitsky

---

## [Author Response · Author response to Decision Letter 0]

1 Jul 2021

Editor

1. election of countries and examples. One suggestion is to focus on a series of countries with better data quality to be able to show in more detail calculations and discussion;

An illustration with good data quality (the Netherlands) from which the values can be computed directly has been added, but I believe it is also important not to shift the discussion too much towards these countries as mortality crises tend to occur in countries with data limitations (or to create situations such that good data collection has to be temporarily suspended).

2. this is related to the presentation of results and discussion. This section could be revised aiming for a more organized and structured presentation. I many parts the reader has some trouble following the main idea

Results and discussion sections have been thoroughly revised.

3. would it be possible to perform sensitive analysis?

Two types of sensitivity analyses have been added. The first one, using data from the Netherlands, analyzes the impact of age grouping (10-year v. 5-year closed intervals and 85 v. 95 as lower bound of open-ended age intervals). The second one, using data from Brazil, analyses the impact of deviation from a “standard” age distribution.

4. improve presentantion of results - there are many issues with the figures that were also reported by reviewers

Results section and figures have been revised.

5. improve the discussion on the limitations of data

The discussion has been added. As pointed out by the 2nd reviewer, however, the emphasis is on the presentation of a new measure. Covid data are used for illustration and the limitations of these data may be different from limitations in other contexts of mortality crisis. 

6. please, see detail comments by both reviewers. 

Reviewer #1: 

1) Why not focusing on a set of countries/population with high quality data at the granularity that is required to conduct the analysis and provide a more in-depth description of these results?

See response to Editor’s point #1 above.

2) The expression […] is confusing to me […], could the author either rephrase this or am I understanding it wrong?

The description has been edited. 

3) Is the [counterfactual calculation of life expectancy in the absence of a cause of death] used by the author in this article analogous to the one described in the cause-deleted section in his co-authored book Preston et al?

The approach is indeed similar, but in Beltran-Sanchez et al. as in textbook, data is on death from all causes from which a counterfactual life expectancy in the absence of a cause is derived; here data (extrapolated from past years) is on death in the absence of a cause to which death from specific cause are added, but underlying assumptions (competing causes of death) are the same in deletion and addition.

4) How is PAYLL calculated? […] in the methods section it says that they come from statistical offices, could you elaborate more on this?

Clarifications with respect to Reviewer 1’s comments #2 and #3 above should also clarify this. PAYLL is a weighted average of counterfactual life expectancies (see 2 above). These are derived from past trends in life expectancies prior to COVID-19 (see 3 above). The latter can be obtained from the UN (for national geographical units) or statistical offices (for sub-national geographical units).

5) A sensitivity analysis on how results differ from 5-year age groups and single year age groups would be useful. Similarly, some discussion on open age interval and how sensible results are to it would inform the reader about limitations

This sensitivity analysis has been added (illustration with data from the Netherlands for using 5-year age groups, and from the Netherlands and Brazil for using a different open age interval).

6) Another approach to the interpolation would be to ungroup cause-specific deaths into single age groups, which would potentially ease the assumption made over n_a_x. This is what is done in the COVerAGE database project (Riffe et al 2021) using a penalize composite link model proposed by Rizzi et al and implemented in the ‘Ungroup’ package in R.

Sensitivity analysis has been limited to the impact of lumping 5-year age groups together. The analysis suggesst the impact is limited. I did not see data that were initially tabulated in 1-year age group, only data that have been modelled in that way, and using those would replace one assumption (made over n_a_x) with another one (involved in modelling one-year age groups from data that may come from 5-year age group). It’s unclear that it would be preferable; its impact on the results would also be more opaque.

7) When calculating the lifetables representing actual mortality conditions (with COVID-19) in each quarter, how did you deal with the exposures? For yearly lifetables we tend to use the mid-year population but for quarterly life tables these should be adjusted.

Yes, these were adjusted by subtracting half of the additional deaths in the current quarter and all of the deaths in the previous quarters (see R script for details).

8) The graphs could be improved by labelling correctly each element

The graphs were edited, labels included.

9) The lack of reproducibility has been noted as an issue in the social sciences. I encourage the author to share data and code, or at least code, used to make the calculations in an open repository. This would not only help in the reviewing process but also encourage readers to use the indicator proposed.

Excel sheets illustrating the calculations and data analyses are added in the supplementary materials. R scripts and all necessary files (except raw files obtained from the UN) have been included in a Github repository.

Reviewer #2: 

1) The main challenge of the paper is the balance between the initial idea-driven sound methodological contribution and the timely and demanding context that the c19 pandemic pose. Honestly, I think the author did not succeed much in balancing the two would be focal topics of the paper. Upon a careful read, I’m convinced that the paper is primarily about the method and not the c19 mortality shock.

The revised paper aims at a better balance, with the emphasis on the new indicator and its interpretation. While COVID-19 was the initial impetus for this, COVID-19 data are used only for illustration purposes.

2) If so, it would be nice to use some well documented mortality shock, for example the heat wave of 2003 in Europe, to illustrate the efficiency of the method on solid and conclusively recorded data. […] It would also allow to mimic the data challenges that arise from the coarseness of the provisional data on c19 deaths – 10y age groups and the open-ended age interval – and get some idea of how much they affect the estimates.

Sensitivity analyses have been added on the coarseness of the COVID-19 data (10-year age groups and open-ended age interval.

3) Results section is very difficult to follow since there is no apparent structure in the way results are presented, and the mentioned in abstract “159 national populations and 122 sub-national populations” are not readily presented.

In order to re-balance the paper toward the methods rather than the results, only four “typical” somewhat comparable populations are presented (Lombardy, New Jersey, Mexico City and Lima), and two more are introduced for the sensitivity analyses (Brazil and the Netherlands).

4) Looking at Fig 3 and the empirical shortcut formula, it seems MUL is limited from the top by PAYLL in the population based on the counterfactual life table. With a sharp spike of deaths, as was seen in Madrid and New York, and the granular time scale, it seems the assumed relevance of the counterfactual life table may be compromised. For example the paper Pifarré i Arolas etal 2021 (#15 in the ref list) clearly shows that the naïve matching of c19 deaths with a reference life table results in a much larger than usual estimate of the average YLL per c19 death.

Indeed, the empirical shortcut suggests that the MUL at most equals the PAYLL (the two being equal when all deaths in a given period are, for example, from COVID-19).

5) [In Figure 1,] the comparison of the quarter estimates within the pandemic year does not provide the support for the appropriateness of using a 2018 life table as a reference.

Figure 1 is not intended to address the appropriateness of using a 2018 life table as reference (in fact, extrapolated 2020 life tables are used as reference). The point of Figure 1 is to show the limited degree to which the distribution of deaths has changed over time in the US. Additional references that appeared since this article was first submitted have been added, as well as a sensitivity analysis on deviations from a “standard” age structure.

6) Under the assumptions of MUL, the paper suggest a translation of p-scores into a YLL measure. This may be more intuitive but inherits the population age structure driven limitations of p-scores.

The fact that the MUL is not an age-standardized measure is duly noted in the paper. What I describe in the “conceptual detour” section of the paper is that the calculation of a new value of life expectancy under mortality crisis conditions removes the effects of the population age structure by assuming stationary conditions. The assumption is not suited for mortality crisis conditions (rapid changes in mortality) and one of the main messages of the paper is that, in my view, for crisis mortality, the “cure” for age-structural dependency is worse than the “disease” so to speak. 

7) Discussion (and also the Introduction) seem to accept the widespread misinterpretation of PLEB as an indicator that (under the hefty assumptions that are mostly omitted in popular interpretations) can inform on the mortality shock effects at the individual level. PLEB is not designed to do so. In a way stating the advantage of MUL (specifically designed to answer this question) against PLEB is not fair.

The introduction and discussion have been revised, hopefully to clarify that I do not accept the widespread misinterpretation of PLEB and that, indeed, MUL is an attempt to derive an indicator that can be interpreted the way most people interpret changes in PLEB. In particular, revised sections aim to clarify the distinction between what I see as (a) a correct use of differences in annual PLEB that stay clear of the above-mentioned widespread misinterpretation, and (b) an incorrect use of what Noymer and Ho have term “pseudo” life expectancies that refer to very short period of time. The advantage of MUL over PLEB for crisis mortality is also, I believe, that it is equally applicable to crises that develop over very What seems a bit confusing in the results is that by MUL the author means only the calculation for one cause of death, namely c19. Since MUL is generic and applicable to any cause of death in any context it would be nice not to build an association between MUL and c19 necessarily.

The revisions aim to emphasize the applicability of MUL to any type of mortality shocks, including those generating deaths from multiple causes (that could always be grouped in “excess” deaths), whereas the illustration is limited to COVID-19, which was also the initial impetus for this undertaking.

8) Much more detailed discussion of c19 data limitations and biases is required. For example, on Line 271 ”but estimates of COVID-19 deaths are available” – under- and untimely reporting happens even in the more developed countries. Karlinsky & Kobak (https://doi.org/10.1101/2021.01.27.21250604) provide an overview of how largely c19 deaths are underreported compared with excess deaths.

As the reviewer notes later, the emphasis of the paper should be, and is now, on the presentation and interpretation of MUL. Data issues that are specific to COVID-19 are duly noted, as this is the example chosen for illustrative purposes here, but should distract from the main aim of the paper. 

9) Similarly, there is no discussion of the limitations imposed by the use of WPP reference life tables that are based on 5x5 Lexis grid and extrapolations.

Correct, this did not appear to be a major contribution to the sensitivity of the results.

10) How exactly were the reference life tables constructed? Were the average 2015–19 WPP life table assumed representing the 2020 counterfactual good enough? A full replication package documenting all the data processing and calculation steps would help a lot.

The 2020 counterfactual was interpolated from the 2015-20 and the 2020-25 WPP life tables. The R scripts have been included in the Github repository.

11) Numbering the equations would improve the readability, cross-referencing, and (in future) citation of the paper

Equations have been numbered.

12) Figures can be improved a lot. Axis titles are missing. All figures will benefit from optimizing the size of the text element – currently too small. Figures 1 and (especially) 2 can benefit a lot from flipping the axes, which would allow to place text horizontally – comfortable for the reader. Figure 1 may be aligned in a population pyramid like fashion. X-axis of figure 3 may be optimized to show some key ticks/dates, e.g. months

All figures have been updated and edited.

13) Some preprints in the references have already been published in peer-reviewed journals, e.g. #15 is now //doi.org/ 10.1038/s41598-021-83040-3.

References have been checked and edited.

14) Minor comments

Line 138: “left” is a bit confusing word in thee context, since these are years lost

Line 227: “because COVID-19 victims are more likely to suffer from other long-term conditions” – a reference is needed, possibly (https://www.medrxiv.org/content/10.1101/2020.10.19.20214494v2)

Line 253: “appears preferable in this case” – reasoning needed

Line 359: “CoViD-19” – written differently

Line 366: “improve MUL estimates by a similarly modest order of magnitude” – I may be mistaken as a non-native English speaker, but I’m used to think that “order of magnitude” refers to a roughly x10 magnitude, which is out of place in the context of the phrase.

Thank you for the reference and careful read. These have been corrected.

---

## [Editor Report · Decision Letter 1]

7 Jul 2021

The Mean Unfulfilled Lifespan (MUL): A new indicator of the impact of mortality shocks on the individual lifespan, with application to mortality reversals induced by COVID-19

PONE-D-21-05770R1

Dear Dr. Heuveline,

We’re pleased to inform you that your manuscript has been judged scientifically suitable for publication and will be formally accepted for publication once it meets all outstanding technical requirements.

Kind regards,

Bernardo Lanza Queiroz, Ph.D

Academic Editor

PLOS ONE

Additional Editor Comments (optional):

I really appreciated the careful revision of the paper. This is an extremely important contribution and making data and codes available will also be very important. 
---

## [Editor Report · Acceptance letter]

16 Jul 2021

PONE-D-21-05770R1 

The Mean Unfulfilled Lifespan (MUL): A new indicator of the impact of mortality shocks on the individual lifespan, with application to mortality reversals induced by COVID-19 

Dear Dr. Heuveline:

I'm pleased to inform you that your manuscript has been deemed suitable for publication in PLOS ONE. Congratulations! Your manuscript is now with our production department. 

Kind regards, 

on behalf of

Dr. Bernardo Lanza Queiroz 

Academic Editor

PLOS ONE